**Data Availability Statement:** The data underlying the results presented in the study are provided in the supplemental data section.

# Fluctuations in quality of life and immune responses during intravenous immunoglobulin infusion cycles

**Jordan K. Abbott**[1¤]*, **Sanny K. Chan**[1,2], **Morgan MacBeth**[1], **James L. Crooks**[2,3], **Cathy Hancock**[1], **Vijaya Knight**[1¤], **Erwin W. Gelfand**[1]

1 Division of Pediatric Allergy-Immunology and the Immunodeficiency Diagnosis and Treatment Program, National Jewish Health, Denver, Colorado, United States of America, 2 Department of Immunology and Genomic Medicine, National Jewish Health, Denver, Colorado, United States of America, 3 Division of Biostatistics and Bioinformatics, National Jewish Health, Denver, Colorado, United States of America

¤ Current address: Department of Pediatrics, Section of Allergy and Immunology, University of Colorado School of Medicine, Denver, Colorado, United States of America

* jordan.abbott@cuasnchutz.edu

## Abstract

Despite adequate infection prophylaxis, variation in self-reported quality of life (QOL) throughout the intravenous immunoglobulin (IVIG) infusion cycle is a widely reported but infrequently studied phenomenon. To better understand this phenomenon, subjects with humoral immunodeficiency receiving replacement doses of IVIG were studied over 3 infusion cycles. Questionnaire data from 6 time points spread over 3 IVIG infusions cycles (infusion day and 7 days after each infusion) were collected in conjunction with monitoring the blood for number of regulatory T-cells (Treg) and levels of 40 secreted analytes: primarily cytokines, chemokines, and growth factors. At day 7, self-reported well-being increased, and self-reported fatigue decreased, reflecting an overall improvement in QOL 7 days after infusion. Over the same period, percentage of Treg cells in the blood increased (p<0.01). Multiple inflammatory chemokine and cytokine levels increased in the blood by 1 hour after infusion (CCL4 (MIP-1b), CCL3 (MIP-1a), CCL2 (MCP-1), TNF-α, granzyme B, IL-10, IL-1RA, IL-8, IL-6, GM-CSF, and IFN-γ). The largest changes in analytes occurred in subjects initiated on IVIG during the study. A significant decrease in IL-25 (IL-17E) following infusion was seen in most intervals among subjects already receiving regular infusions prior to study entry. These findings reveal several short-term effects of IVIG given in replacement doses to patients with humoral immunodeficiency: QOL consistently improves in the first week of infusion, levels of a collection of monocyte-associated cytokines increase immediately after infusion whereas IL-25 levels decrease, and Treg levels increase. Moreover, patients that are new to IVIG experience more significant fluctuations in cytokine levels than those receiving it regularly.

**Funding:** This work was carried out under an unrestricted grant from CSL Behring Inc. SKC, JKA and EWG. CSL Behring Inc. played no role in the study design, data collection, analysis, or manuscript preparation. The manuscript was reviewed but not edited by CSL Behring prior to submission. JKA receives funding from NIH K08AI141734. The funders had no role in study design, data collection and analysis, decision to publish, or preparation of the manuscript.

**Competing interests:** The authors have declared that no competing interests exist.

## Introduction

Intravenous Immunoglobulin (IVIG) replacement prevents infection in primary immunodeficiency diseases (PIDD) with defects in antibody production [1] and is used as an immuno-modulatory and anti-inflammatory therapy in a variety of diseases [2]. Particularly in patients receiving IVIG for PIDD, it has been widely observed that patients experience a phenomenon of "wear-off" in quality of life or feelings of well-being following infusion [3]. Wear-off occurs in patients receiving IVIG every 3 to 4 weeks and is characterized by symptomatic improvement following infusion with subsequent deterioration at some time preceding the subsequent infusion, typically the preceding week.

Varied immune parameters are also known to fluctuate in response to IVIG. Transient increases in blood Treg levels and transient decreases in blood inflammatory monocyte levels have been observed in the immediate post-infusion period [4,5]. Longer-term changes in other immune lineages have also been associated with the initiation and continued use of IVIG infusion in immunoglobulin deficient populations [6]. Additionally, chemokine and cytokine levels are known to fluctuate following IVIG infusion [7]. These shifts suggest a dramatic interplay between IVIG and the immune system, one that provides rationale for its use in a variety of immunodeficient and immunodysregulatory conditions.

The implications of these transient immunologic alterations to clinical parameters are not known. Since the underlying mechanism of IVIG wear-off is also poorly understood, we sought to better understand the changes in immunologic parameters that occur over the time frame during which QOL changes. This study was initiated to characterize changes in either Treg numbers and/or blood cytokine and chemokine levels that occur around the times when IVIG wear off are typically reported. We show that IVIG wear-off is a common phenomenon among immunodeficient patients, and both Tregs and numerous analytes fluctuate in amount over the time-period when IVIG-wear off occurs. Novel features of these shifts in immune parameters are presented.

## Materials and methods

### Study design

Subjects were monitored over the course of 3 infusion cycles beginning with a visit on the day of the infusion and followed by a visit 7 days after the infusion (day 7). In total, there were 6 study visits. Blood was drawn on infusion days both before (pre-infusion) and 1 hour after infusion completion (post-infusion). On day 7, blood was drawn once. Questionnaires were administered on the day of infusion and on day 7 (**Fig 1**). This sequence was repeated 3 times for a total of 3 cycles.

### Subjects

Subject data are presented in **Table 1**. All subjects signed an IRB-approved consent form to participate in the study. The inclusion criteria allowed for enrollment of adult patients receiving IVIG every 4 weeks for humoral immunodeficiency. Patients on other immunomodulatory medications were excluded. Twenty subjects met inclusion and exclusion criteria. Two subjects were enrolled but were lost to follow up prior to completing any infusion day to day 7 intervals. The diagnoses included: combined immunodeficiency (n = 2), common variable immunodeficiency (CVID, n = 10), X-linked agammaglobulinemia (XLA, n = 3), and hypo-gammaglobulinemia (n = 3). In total, eighteen subjects with a clinical and laboratory diagnosis requiring replacement immunoglobulin therapy were enrolled and completed the study with at least 2 evaluable intervals: 15 subjects completed 6 visits, 1 subject completed 5 visits, and 2

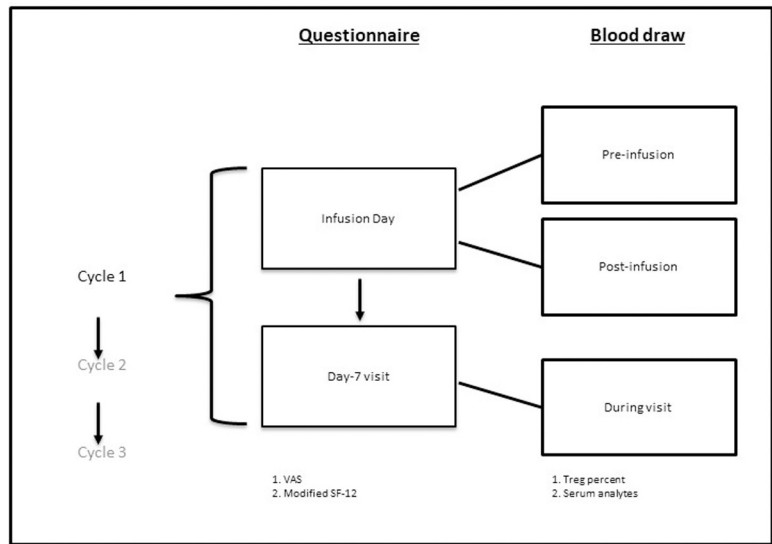

**Fig 1. Outline of study visits.** Subjects participated in three identical cycles. Each cycle consisted of an infusion-day visit and a day-7 visit (2 visits per cycle). Questionnaires were completed once per study visit day for a total of 6 time points. Blood was drawn for Treg levels and blood analytes 3 times per cycle, twice on the infusion day (before infusion and 1 hour after completion) and once at the day-7 visit.

subjects completed 4 visits. The mean age at enrollment was 50.1 years ± 17.8 years. Twelve subjects (66.7%) were female. The mean IVIG replacement dose was 494 ± 110 mg/kg. All subjects received IVIG at 4-week intervals through the duration of the study. Three subjects initiated IVIG during the study (naïve group, mean age 66.3 ± 10.5 years, all female). Twelve subjects received Gamunex-C (Grifols) and 6 subjects received Privigen (CSL Behring).

**Table 1. Patient demographics.**

| Subject ID | Age (yr) | IVIG Indication | Sex | IVIG Dose (g/kg) | IVIG Naïve? | # Study Visits |
|---|---|---|---|---|---|---|
| 01 | 67 | Hypogammaglobulinemia | Female | 0.42 | no | 7 |
| 02 | 52 | Combined immunodeficiency | Male | 0.59 | no | 7 |
| 03 | 72 | Hypogammaglobulinemia | Female | 0.36 | no | 7 |
| 04 | 61 | XLA | Male | 0.63 | no | 7 |
| 05 | 18 | Combined immunodeficiency | Female | 0.73 | no | 4 |
| 06 | 51 | CVID | Female | 0.48 | no | 7 |
| 08 | 48 | CVID | Male | 0.47 | no | 6 |
| 10 | 68 | CVID | Female | 0.39 | no | 7 |
| 11 | 52 | CVID | Female | 0.37 | yes | 6 |
| 12 | 60 | CVID | Female | 0.51 | no | 7 |
| 13 | 33 | XLA | Male | 0.50 | no | 7 |
| 14 | 32 | CVID | Female | 0.72 | no | 7 |
| 15 | 31 | CVID | Female | 0.39 | no | 7 |
| 16 | 33 | CVID | Male | 0.40 | no | 7 |
| 17 | 20 | XLA | Male | 0.54 | no | 5 |
| 18 | 77 | CVID | Female | 0.38 | yes | 7 |
| 19 | 57 | CVID | Female | 0.53 | no | 6 |
| 20 | 70 | Hypogammaglobulinemia | Female | 0.50 | yes | 4 |

## Questionnaires

Subjects were administered two QOL questionnaires on the infusion day and day 7: a modified SF-12-style instrument that was adjusted to better capture perceived health over the week prior to administration of the questionnaire and a visual analog scale (VAS) (**S1 Fig**). The modified SF-12-style instrument consisted of 12 total questions with Likert scale responses between 1 and 5. The VAS is a single question where subjects rate their health on a scale of 0 to 100. Questionnaires were administered once on each visit day.

## Measurement of T$_{reg}$ levels

Peripheral blood mononuclear cells (PBMC) were separated from heparinized whole blood using density gradient centrifugation and incubated with 20 mcL each of CD4-FITC, CD25-PE and CD3-PerCP-Cy5.5 (all purchased from BD Biosciences, San Jose, CA) for 20 minutes. The cells were permeabilized and stained with 20 mcL of anti-FoxP3-Alexa647 (BD Biosciences) for 30 minutes. Regulatory T cells were identified as CD3+CD4+CD25$^{hi}$FoxP3 + lymphocytes, gating initially on lymphocytes based on forward- and side-scatter, and were reported as a percentage of CD3+CD4+ T cells.

## Measurement of blood cytokines and chemokines

Serum analytes were measured using the Luminex Human XL Cytokine Discovery Kit (R&D Systems) run on the MAGPIX CCD imaging system (Luminex Corp.). Standard curves were generated using a 5-point weighted logistic model using xPONENT 4.2 software. For the mixed model analysis, values below the limit of detection (LOD) were removed. For the remainder of analysis, the lower of either the pre-programmed LOD/2 or lowest measured value/2 were used.

## Statistical analysis

All analyses were performed in the R language (version 3.6.0) [8]. For ordinal questionnaire data, relationships between infusion day and day 7 pre-infusion were analyzed using the clmm function from the ordinal package (version 2019.4–25) [9] with random intercept by subject, probit link, and equidistant cut-points, though similar results were found with the lmer function with Gaussian response. VAS score and T$_{reg}$ numbers were analyzed using the lmer function from the lme4 package (version 1.1–21) [10] with random intercept by subject. Log-transformed cytokine and chemokine concentrations were analyzed using the lmec function in the lmec package (version 1.0) [11] with random intercept by subject and left censoring of concentrations below the assay limit of detection (LOD). The Benjamini-Hochberg procedure was used to adjust p-values to control the rate of false discoveries among the cytokines and chemokines [12]. Figures were produced with the following R packages: ggplot2, corrplot, ggpubr, and tidyverse.

In post-hoc analysis, VAS score was normalized to the subject mean. Kruskal-Wallis test was used to assess correlation between VAS and SF-12 responses. For cytokine analysis, concentrations below the LOD were substituted with the lesser of LOD/2 or the lowest value reported divided by 2 even if lower than the reported LOD, which varied by cytokine/chemokine. Cytokine and chemokine concentrations were log-transformed prior to analysis. Influence of patient features on cytokine changes relative to infusion were analyzed by ANOVA.

**Table 2. Significant relationships between QOL measures on infusion day and day 7.**

| Measure | Coefficients | p-values | CI_2.5 | CI_97.5 |
|---|---|---|---|---|
| Weekly Health (1–5 scale, 1 is high) | -0.5088 | 0.0295 | -0.9552 | -0.0624 |
| Suffer from fatigue? | -0.4840 | 0.0320 | -0.9154 | -0.0527 |
| VAS | 4.6440 | 0.0003 | 2.1439 | 7.1441 |

## Results

### Self-reported QOL increases following IVIG infusion

Self-reported quality of life was assessed on the day of infusion—the presumed QOL nadir—and at day 7, when the positive effect of IVIG was likely to have onset and unlikely to have diminished (Fig 1). Changes over this interval were modeled with random intercept by subject to control for inter-subject variation. Under this model, the overall VAS score increased (improved) at day 7 (p = 0.0003, Table 2, Fig 2A). In the modified SF-12 questionnaire, fatigue (question 5) and general assessment of health for the prior week (question 2) decreased (improved) over this same period (p = 0.03, p = 0.03, Table 2, Fig 2C & 2D). Responses to question 2 support the idea of improved health following infusion. A score of 4 –indicating current health worse than 1 week prior–occurred on the infusion day 3 times as frequently as on day 7 (n = 9 vs n = 3). Of those 9 times, the score decreased on the subsequent day 7 visit

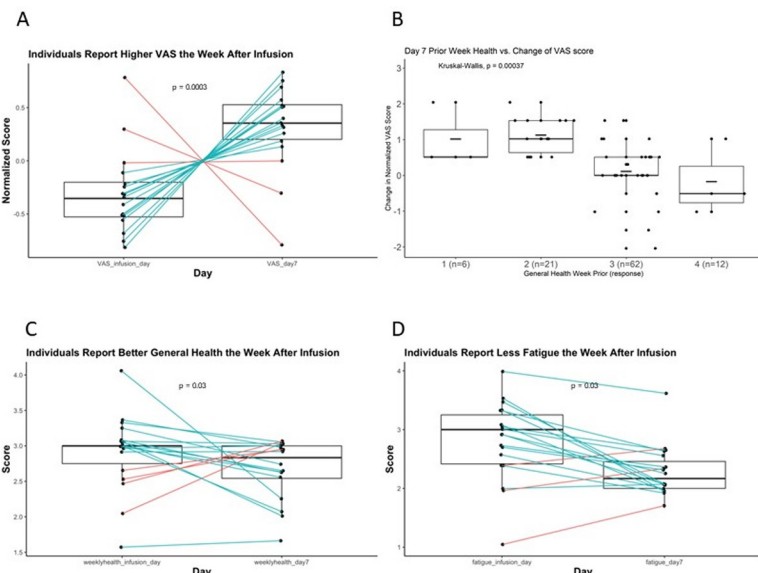

**Fig 2. Generalizable improvement of QOL measures on day 7 vs. infusion day. . A.** Paired boxplot representing the change in normalized VAS score from infusion day to the day-7 visit. The box borders represent the first and third quartiles with the whiskers extending to 1.5*(interquartile range). Median is represented by a horizonal line. VAS scores were normalized to Z-scores by subtracting the mean for each subject and dividing by the standard deviation. **B.** Boxplot of response to question 2 from the modified SF-12: "*Compared to the week before, how would you rate your health in general now?*" vs. change in normalized VAS score between infusion day and day 7. Box features as in 2A with mean scores are marked by a short horizontal bar. Intervals during which patients reported no improvement (#3) or worsening (#4) of general health had lower normalized VAS scores at day 7 and patients who reported improvement (#1 or #2) had higher normalized VAS scores. Responses: "1" = 3, "2" = 14. "3" = 30, "4" = 3. **C.** Paired boxplot representing change in weekly-health score (question 2) between infusion day and day 7. Scores were normalized to the mean for each subject. **D.** Paired boxplot representing change in fatigue scores (question 5) between infusion day and day 7. Fatigue scores were normalized to the mean for each subject. Infusion day scores are enriched for positive values (more fatigue) and day 7 scores are enriched for negative scores (less fatigue).

on 7 occasions and remained at 4 on 2 occasions (**S2C Fig**). We then tested for correlations between the VAS and individual items from the modified SF-12. Changes in VAS score were significantly correlated with modified SF-12 question 2 (health for the prior week) on day 7 (Kruskal Wallis, p = 0.001, **Fig 2B**) but did not correlate with question 5 (fatigue) either at day 7 or as the difference in fatigue scores between infusion day and day 7 (p = 0.13, p = 0.32, **S2A Fig**). In fact, 2 of the 3 subjects who on average reported lower VAS scores on day 7 reported an improvement in fatigue on day 7, whereas 2 of the 3 subjects who on average reported more fatigue on day 7 reported higher VAS scores at the same time point. These findings support and provide insight into IVIG "wear-off". VAS scores were consistently lower for most subjects on the day of infusion compared to 7 days later. Rating of weekly health over the past week likewise improved on day 7. Fatigue over the preceding week decreased from infusion day to day 7; however, fatigue scores did not correlate with the VAS score or day 7 report of prior weeks. Therefore, the questionnaires captured two independent quality of life measures–general health and fatigue–that changed from infusion day to day 7.

Changes in QOL measures were evaluated for association with demographic factors and other features. There was no subgroup difference in change of VAS, day 7 weekly health score, change in weekly health score, or change in fatigue score when the cohort was compared by sex, diagnosis, serum IgA level, serum IgM level, IVIG dose, infusion pre-treatment or IVIG brand. When analyzed in isolation, change in VAS correlated with subject age (**S2B Fig**); however, when controlling for other factors, the p-value was no longer statisticaly significant. There was no correlation between age and day 7 weekly health score, change in weekly health score or change in fatigue score. Therefore, change in VAS score was not associated with demographic features of the study group; however, there is a trend towards older subjects reporting larger changes in VAS.

## Treg numbers increase following IVIG infusion

Treg numbers were measured at 3 time points during the cycle: before infusion, one hour after infusion completion, and 7 days after the infusion. There were no significant changes in numbers between the pre-infusion and post-infusion blood draw; however, levels increased at the visit 7 days following the infusion (p = 0.01, **Figs 3** and **S3A**). There was substantial variability among individuals from cycle to cycle. From before infusion to day 7, five subjects experienced an increase in Treg percentage every cycle, 8 experienced an increase on 2 cycles, and 5 experienced an increase during only 1 cycle. All subjects experienced a Treg increase during at least 1 cycle. We tested whether Treg levels on infusion day, day 7, or change from pre-infusion to day 7 correlated with changes in VAS score or fatigue score over the same time interval. There was no correlation between Treg levels and changes in VAS score or changes in fatigue score (**S3B & S3C Fig**), suggesting either the absence of a relationship between increases in Treg numbers and the assessed QOL measures, or insufficient numbers of measurements to identify such a relationship.

The subjects were divided based on demographic and other features and analyzed for differences in the change of Tregs percentage from infusion day to day7. There were no associations between sex, IVIG product, diagnosis, age, IVIG dose, baseline IgA level, baseline IgM level, infusion pre-treatment, or naive status and magnitude of change in Treg percentage.

## Changes in serum cytokine and chemokine levels during the infusion cycle

Multiple changes in cytokine and chemokine levels occurred between the pre- and 1h post-infusion blood draws. CCL2, CCL3, CCL4, TNF-α, granzyme B, IL-10, IL-1RA, IFN-γ, IL-8, and CCL20 increased while IL-25 decreased (**S8 Fig**). After stratification of naïve vs. IVIG-

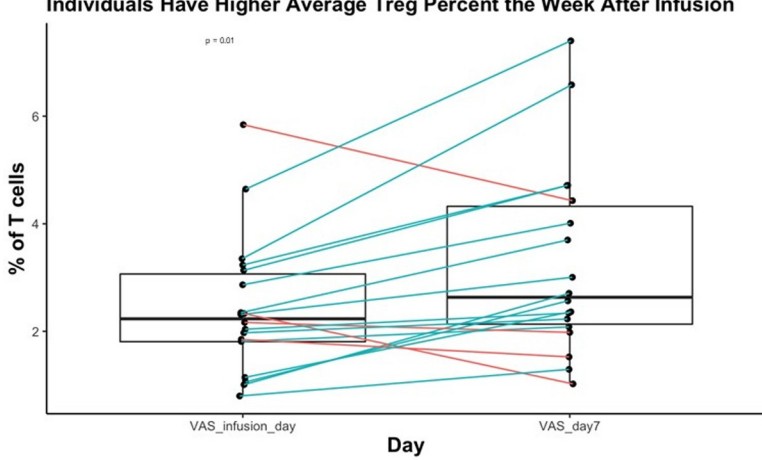

**Fig 3. Proportion of T cells with Treg Phenotype increases from pre-infusion to day 7.** Paired boxplot representing the change in Tregs as a percentage of CD4+ T cells from the pre-infusion blood draw to day 7. Treg% was averaged for each subject before infusion and on day 7. P-value is derived from the random effects model of the relationship between day 7 and the pre-infusion draw.

experienced subjects, (**Fig 4**) additional significant increases in IL-6 and GM-CSF and significant decreases in EGF and CD40L were seen in IVIG-naïve subjects. In IVIG-experienced subjects, CCL3, CCL4, and granzyme B were the only significant cytokine increases, with increases in CCL2, TNF-α, IL-10, IL-1RA, IFN-γ, IL-8, and CCL20 no longer attaining significance. Interestingly, the decrease in IL-25 level identified in the non-stratified subject sample was entirely attributable to the non-naïve group and was not detected in IVIG-naïve subjects (**Fig 4**). Thus, IVIG infusion in IVIG-naïve subjects is rapidly followed by an increase in a subset of inflammatory cytokines in the blood. In IVIG-experienced subjects, a partially

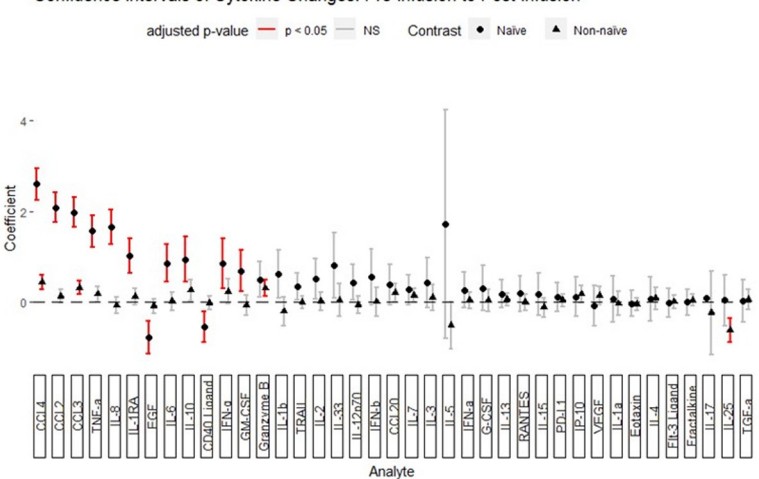

**Fig 4. Statistically-significant cytokine changes between the pre-infusion and post-infusion blood draws plot of the mean (dot) and confidence interval (whiskers) for the coefficients of the pre-infusion to post-infusion interval.** Subjects were separated by whether IVIG treatment was initiated during the study (naïve) or if the subject was already receiving IVIG regularly (non-naïve). P-values for coefficients were adjusted by the Benjamini-Hochberg method. Adjusted p-values < 0.05 are highlighted in red.

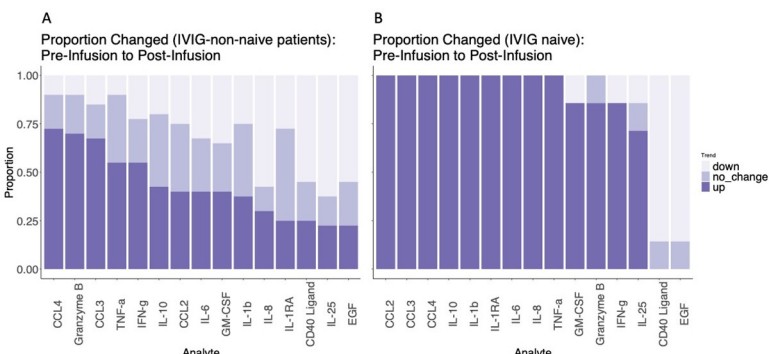

**Fig 5. Proportions of intervals where cytokines change varies between cytokines.** Stacked bar graph for each cytokine found to change between the pre- and post-infusion measurements; A. non-naïve subjects, B. naïve subjects. Direction of change is signified by depth of color as shown in the figure legend. "Up" signifies a greater than 10% increase and "down" signifies a greater than 10% decrease.

overlapping set of changes is identified with similar but lower magnitude changes in CCL3, CCL4, and granzyme B, and a distinct decrease in IL-25 that was not detected in the naïve subjects.

To determine whether these changes represented large changes in few subjects or whether these changes were widespread, a direction of change was assigned for each interval and the proportions of direction of change were examined (**Fig 5**). CCL3, CCL4, granzyme B, TNF-α, and IFN-γ *increased* for the majority of pre-to-post infusion intervals, whereas IL-10, CCL2, IL-6, GM-CSF, IL-8, and IL-1RA were increased in only the minority. Among the cytokines found to decrease significantly, CD40 ligand, IL-25, and EGF *decreased* for most pre-to-post infusion intervals (**Figs 5, S4, S5 and S6**). These findings suggest that increases in the levels of CCL3, CCL4, granzyme B, TNF-α, and IFN-γ are typical changes in response to IVIG infusion, as are decreases in CD40 ligand, IL-25, and EGF.

We then evaluated whether demographic and clinical features differentially associated with pre- to post-infusion cytokine changes that were significantly different in the group as a whole. Univariate analysis identified several trends related to diagnosis, age, IVIG dose, baseline IgA level, baseline IgM level, infusion pre-treatment, and naive status (**S9–S18 Figs**). Multivariate ANOVA confirmed that naive status was significantly associated with post-infusion increases in CCL2, CCL3, CCL4, IL-1RA, IL-8, and TNF-α. Baseline IgA level was directly associated with increases in CCL4 and IL-10 levels. Age was associated with larger increases in IL-10 levels. Diagnosis was associated with variation in the IL-10 level. Lower IVIG dose was associated greater increases in the IL-8 level. This indicates that patients naive to IVIG have greater increases in monocyte-related cytokines than those have been receiving it for some time. In addition, diagnostic category, baseline IgA level, and IVIG dose exert an influence on changes in cytokines following IVIG infusion, albeit in this cohort the effects appear to be limited in scope.

Comparing pre-infusion to day 7, no changes in cytokine levels reached statistical significance of adjusted p-value < 0.05; however, IL-25 trended towards a decrease during the interval (p = 0.051). As IL-25 was decreased immediately after infusion, we further determined that the log-fold change in IL-25 between the pre-infusion and post-infusion time point was not different than the pre-infusion and day 7 time points (paired T-test, p = 0.46, **S7 Fig**). Therefore, in the IVIG-experienced group, IL-25 levels decrease immediately following IVIG infusion and frequently remain low at least 7 days post-infusion. IL-25 decreases occurred in 55% of pre-to-post intervals and 58% of pre-to-day-7 intervals and increases occurred in 30% of

both intervals. Thus, IVIG infusions are frequently associated with immediate and prolonged decreases in IL-25 in a subset of subjects.

## Correlation analysis links clusters of cytokines

To better understand the relationship between cytokines, we calculated correlation coefficients for changes in cytokine/chemokine levels before and immediately after infusion. In IVIG-experienced subjects, CCL2, CCL3, CCL4, TNFα, IL-8, IL-1RA, granzyme B, and IL10 were highly correlated (**Fig 6**). This suggests that these cytokines were secreted by the same cell or at least were induced in response to the same stimulus. IL-25 was not correlated with this group of cytokines, indicating that decreases in IL-25 occurred independently of increases in the identified inflammatory cytokines and chemokines.

## Relationships between cytokine changes and QOL measures

Changes in cytokine levels were tested for relationships with either the change in VAS or fatigue score from infusion day to day 7. IVIG-naïve subjects were not evaluated independently as the number of measurements was inadequate to power such an analysis. No linear relationship was seen between change in VAS or change in fatigue score and any of the cytokines or chemokines that were significantly changed in the study.

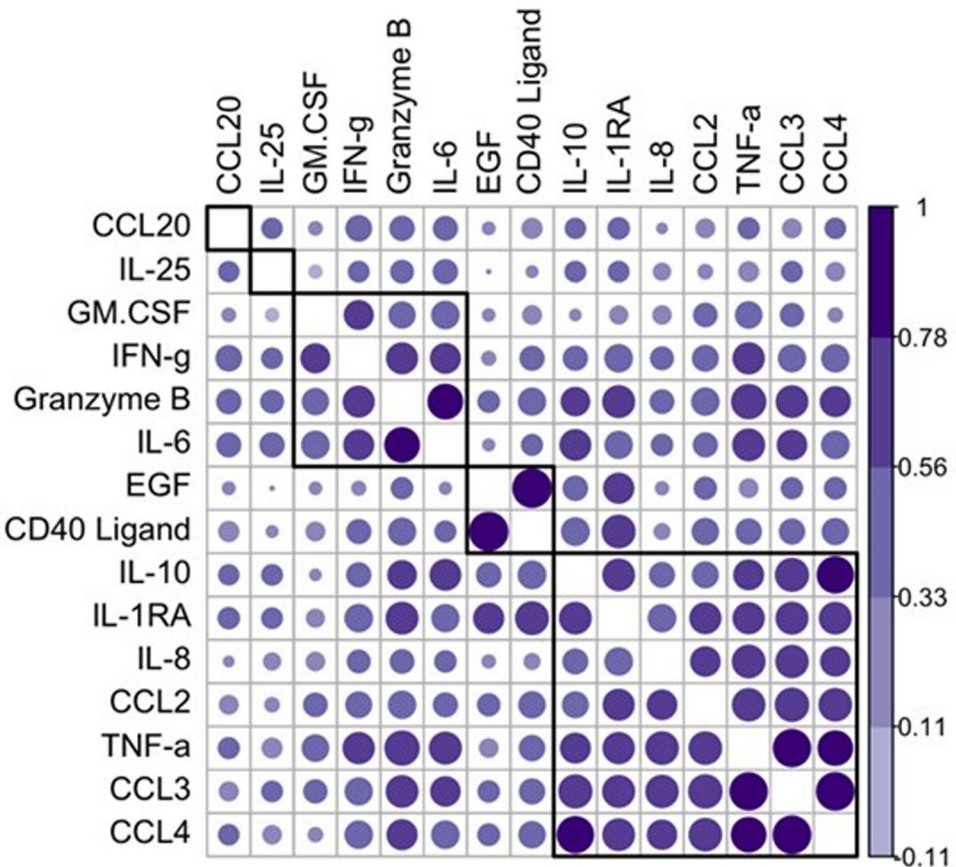

**Fig 6. Correlations of cytokines that significantly change between the pre-infusion and post-infusion blood draws pearson's correlation between cytokine pairs is indicated by color and size of dots.** The larger and deeper colored dots indicate a higher r value. The analytes were hierarchically clustered, and the black boxes correspond to clusters generated at a level of 6 clusters.

## Discussion

The objective of this study was to better characterize QOL and serum analyte fluctuations during the IVIG cycle. Despite the limited sample size, we identified several significant shifts in the targeted measures. We confirmed that following infusion of replacement doses of IVIG there is a decline in quality of life in a subset of patients over time that is reversed with the subsequent infusion. We showed that Treg levels increase over 7 days following IVIG infusion. We showed that several serum cytokines and chemokine levels change immediately and at the day 7 time point. We performed an analysis to determine if these blood changes correlated with changes in QOL measures but were unable to identify any clear relationship. In this patient cohort, IVIG improved measures of general health and fatigue while simultaneously affecting Treg, chemokine, and cytokine levels, but the relationship between these cytokine changes and self-perceived quality of life remains to defined.

The "wear off" phenomenon has been reported in multiple contexts [3,13,14], and patients with a primary immune deficiency disease often report a significant diminished quality of life compared to their "normal" counterparts. In this study, 50% of all intervals between day 7 and the next infusion were associated with a decrease in VAS score. These intervals where worsening occurred involved a majority of patients (12 of 17, 70%) with evaluable intervals. Rojavin et al. reported "wear off" in 43% of patients on a 4-week cycle but only in 10% of all cycles [3]. Our approach was different in that they analyzed QOL changes between week 2 and the last week of dosing after decreases in QOL occurs which may account for some of the differences in the two studies. Another important difference with the Rojavin study is the higher mean age of the current study group, a demographic factor that trends with variation in the VAS score. Additional variation could arise from other differences in the patient population, length of IVIG treatment, and geography.

The impact of IVIG infusion on $T_{reg}$ cells has been studied extensively in the context of inflammatory disease and less so in primary immunodeficiency. The proportion of Tregs increased following high-dose IVIG-treatment in patients with vasculitis [15,16], Guillain-Barre syndrome [17], and Kawasaki disease [18]. Likewise, a prior report indicated an increased percentage of CD4 $T_{reg}$ 30 minutes following replacement-dose IVIG infusion in CVID patients [4]. While we did not confirm that $T_{reg}$ levels as a percentage of CD4 T-cells increase immediately following infusion, we saw a significant increase at the day 7 in patients receiving replacement-dose IVIG. This change in Treg proportion did not appear to be associated with QOL changes over the measured interval, suggesting that in this cohort, Tregs were not directly involved in the process that drives improved QOL over the week following infusion.

This study aimed to identify associations between cytokine changes and variation in QOL through the IVIG cycle. Many of the cytokines monitored have been implicated in altering CNS function and mood, and as a result, may affect QOL determinations [19]. Despite identifying quite large immediate cytokines changes, we did not find associations between these changes and the reported QOL improvements. There are multiple likely reasons for this. First, the non-naive group, which made up most of the study population, experienced lower magnitude and less significant cytokine level changes than the naive group, so there were less dramatic changes available for comparison. Second, the low number of enrolled subjects restricts the power of the study to identify subtle correlations missed here. Third, it is possible that timing of this relationship differs from that designed in the study in that additional analyte changes may be occurring between the infusion and day 7. Fourth, the relationship between cytokine fluctuations in the blood and alterations in the CSF is not entirely clear. It is conceivable that changes in the CNS such as blood-brain barrier alterations, increases or decreases in

glymphatic flow, or changes in tissue cytokine levels may have minimal impact on serum levels, particularly when dramatic shifts in analyte levels seem to be simultaneously occurring in other parts of the body. Given the demonstrated role of LPS in altering CNS glymphatic function (20), the known role of low level chronic LPS exposure in CVID [20], and the ability of IVIG to reduce LPS levels in the blood, a corresponding change in CNS conditions may be expected in patients that suffer from chronic LPS exposure. Considering these aspects, it is difficult to rule out that cytokine changes directly or indirectly induced in the CNS do in fact correlate with the reported QOL changes. Innovative approaches to monitoring CNS physiology would be helpful to further understand these relationships.

Despite not finding cytokine-QOL associations, this study identified important and potentially relevant changes in cytokine levels that follow IVIG infusion in patients with humoral immunodeficiency. It has previously been reported that inflammatory cytokines such as TNF-α, IL-6, IL-1RA, and IL-8 levels increased following IVIG infusion in similar patient populations [7,21]; however, we have shown that these changes predominate in IVIG-naïve subjects, suggesting an adaptation in experienced subjects that diminishes these responses. At least part of this process remains active in IVIG-experienced subjects, as evidenced by the statistically significant increases in CCL3, CCL4, and granzyme B that persist, albeit at a lower level. The timing of this adaptation has not yet been studied but is likely greater than 3 months, as dramatic shifts in cytokines were still observable during the third infusion in the IVIG-naïve group. Another novel finding was the decrease in serum IL-25 level detected both after infusion and 7 days later in a subset of IVIG-experienced subjects. Decrease in IL-25 following infusion trended with increasing age of the subject. IL-25, among other functions, amplifies atopic inflammation and is produced by various cell types including epithelial cells, Th2 cells, alveolar macrophages, mast cells, basophils, and eosinophils can produce IL-25 [22].

In summary, in this study, the QOL "wear off effect" following IVIG infusion was confirmed, and although several potentially relevant factors were identified, a clear role for these factors in this effect remain elusive. In this cohort, increases in Treg levels and various serum chemokines and cytokines did not correlate with reported improvement and subsequent deterioration in QOL throughout the IVIG cycle. Nonetheless, novel findings regarding replacement-dose IVIG infusions were discovered particularly relating to the differential response of IVIG-naïve vs. IVIG-experienced subjects. This constellation of findings provides a framework for future work exploring the non-infectious physiologic alterations induced by IVIG infusions and how they relate to the feeling of well-being among the highly burdened PIDD patient population.

## Supporting information

**S1 Fig. Modified SF-12 and visual analog scale.**
(PDF)

**S2 Fig.**
(PPTX)

**S3 Fig.**
(PPTX)

**S4 Fig.**
(PPTX)

**S5 Fig.**
(PPTX)

**S6 Fig.**
(PPTX)

**S7 Fig.**
(PPTX)

**S8 Fig.**
(PPTX)

**S9 Fig.**
(PPTX)

**S10 Fig.**
(PPTX)

**S11 Fig.**
(PPTX)

**S12 Fig.**
(PPTX)

**S13 Fig.**
(PPTX)

**S14 Fig.**
(PPTX)

**S15 Fig.**
(PPTX)

**S16 Fig.**
(PPTX)

**S17 Fig.**
(PPTX)

**S18 Fig.**
(PPTX)

**S1 Data.**
(XLSX)

## Acknowledgments

We are grateful to the patients who participated in the study.

## Author Contributions

**Conceptualization:** Jordan K. Abbott, Vijaya Knight, Erwin W. Gelfand.

**Data curation:** Jordan K. Abbott, Morgan MacBeth, James L. Crooks, Vijaya Knight.

**Formal analysis:** Jordan K. Abbott, Morgan MacBeth, James L. Crooks.

**Funding acquisition:** Jordan K. Abbott, Erwin W. Gelfand.

**Investigation:** Jordan K. Abbott, Vijaya Knight, Erwin W. Gelfand.

**Methodology:** Jordan K. Abbott, Morgan MacBeth, Vijaya Knight, Erwin W. Gelfand.

**Project administration:** Jordan K. Abbott, Sanny K. Chan, Cathy Hancock, Erwin W. Gelfand.

**Supervision:** Jordan K. Abbott, Sanny K. Chan, Cathy Hancock, Erwin W. Gelfand.

**Visualization:** Jordan K. Abbott.

**Writing – original draft:** Jordan K. Abbott, Erwin W. Gelfand.

**Writing – review & editing:** Jordan K. Abbott, Sanny K. Chan, Vijaya Knight, Erwin W. Gelfand.

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
