## [Decision Letter · Decision Letter 0]

2 Dec 2021

PONE-D-21-33136Fluctuations in quality of life and immune responses during intravenous immunoglobulin infusion cyclesPLOS ONE

Dear Dr. Abbott,

Thank you for submitting your manuscript to PLOS ONE. After careful consideration, we feel that it has merit but does not fully meet PLOS ONE’s publication criteria as it currently stands. Therefore, we invite you to submit a revised version of the manuscript that addresses the points raised during the review process.

Both the reviewers have commented that the article is of interest to the community. Additional data and interpretation are required.

We look forward to receiving your revised manuscript.

Kind regards,

Jagadeesh Bayry, DVM, PhD, HDR

Academic Editor

PLOS ONE

Journal Requirements:

Reviewers' comments:

Reviewer's Responses to Questions

**Comments to the Author**

1. Is the manuscript technically sound, and do the data support the conclusions?

Reviewer #1: Yes

Reviewer #2: Partly

2. Has the statistical analysis been performed appropriately and rigorously? 

Reviewer #1: Yes

Reviewer #2: I Don't Know

3. Have the authors made all data underlying the findings in their manuscript fully available?

Reviewer #1: Yes

Reviewer #2: Yes

4. Is the manuscript presented in an intelligible fashion and written in standard English?

Reviewer #1: Yes

Reviewer #2: Yes

5. Review Comments to the Author

Reviewer #1: The paper is of some interest, however several aspects need to be clarified. QOL is the results of a number of factors, including psycological, behavioural and social factors. Moreover, the authors tested a number of cytokines that have a role in immunity, but their role on brain, i.e. on mood is less clear.

The authors should determine whether the change in QOL is a true change (and not linked to a positive or negative reinforcement due to the administration of IVIg per se).

It is likely that a depression questonnaire could be helpful.

Also, the authors should consider how the administration of IVIg leads to the apparent feeling or freedom and safety thus allowing 8and leading into) a better QOL.

Finally, all the immunologic changes are somehow expected, but how do they impact on patients moddo and functionality? any relationship with psychological behaviours?

Reviewer #2: This is an interesting paper, but a bit more data and interpretation could help the reader assess the applicability and clinical significance of the conclusions to individual patients. The data and comments requested below should be available already, but should be added to the presentation. It is probably not feasible at this time, but it would have been informative to have weekly data points throughout each infusion-infusion cycle, rather than just comparing the pre-infusion and 7-day time points.

Answers to a few questions would help the readers focus on the results:

1. Were pre-infusion meds such as NSAIDS, aspirin or steroids allowed, or used in any patients ? Were there any clinically significant infusion reactions and did these lead to results different from infusions with no reactions ?

2.Were any patients on continuous prophylactic or treatment antibiotics ?

3. Please add to table 1 data on the duration of IVIG treatment and/or number of infusions preceding enrollment for each subject.

4. The authors should clarify if their use of the term "VAS" score refers only to their 100 point "thermometer" and not to the 5 point boxed choices on the modified SF-12.

5. In fig 2B, the authors should indicate the number of values for each response category (1-4), indicate whether any subject ever reported a value of 5, and comment on the consistency of individual subjects' responses over the multiple intervals represented in the figure.

To aid in interpreting the results, the authors should discuss whether there is any data on the clinical relevance of the change in Tregs from 2.28 % to 2.42% portrayed in Fig 3, and discuss the minimal clinically important difference in this measurement, as well as in the cytokine values in fig 4. Is there any data available on the range of results for normal controls in their laboratory ? In addition, different symbols or colors should be used for IVIG-naive vs experienced subjects. The authors should also discuss the degree of variability vs consistency in Treg responses in different infusion cycles in individual subjects (fig S3, and S4-S5)

Fig 5 could also be split into two figures- one showing the IVIG-naive subjects and one showing the experienced subjects.

The authors conclude that the changes in IL-25 they report may partially explain "the clinical efficacy of IVIG in allergic disease", citing the over-20 year-old reference #21 in regard to this "efficacy". What is the evidence that the decrement in IL-25 they report is clinically significant? Given the lack of substantiation of the efficacy of IVIG in most atopic diseases, and the facts that the FDA has not approved IVIG for any atopic disease, and that the AAAAI review of uses of IVIG does not recommend it for atopic disease, this conclusion appears to be an overstatement rather than a "take-home message" as it seems currently stated. The authors should probably substitute the latest AAAAI recommendations for IgG therapy (Perez et al JACI 2017) for the outdated reference 1 they now cite.

6. PLOS authors have the option to publish the peer review history of their article (what does this mean?). If published, this will include your full peer review and any attached files.

Reviewer #1: No

Reviewer #2: No

---

## [Author Response · Author response to Decision Letter 0]

4 Feb 2022

note: response is also provided as a word document with formatting.

1/19/2022

To the Reviewers,

Thank you for taking the time and attention to review our submitted manuscript. The comments were insightful, and by addressing your concerns, we feel that the quality of the manuscript has significantly improved. A point-by-point response to the concerns is listed below. We hope that you find that our responses adequately address your concerns and answer your questions.

Sincerely,

Jordan Abbott, MD, MA

University of Colorado, School of Medicine

Department of Pediatrics, Section of Allergy and Immunology

Note for the Editor:

CSL Inc. has granted permission to supply the deidentified dataset. It is included as an excel file labeled supplemental data.

Reviewer #1: The paper is of some interest, however several aspects need to be clarified. QOL is the results of a number of factors, including psycological, behavioural and social factors. Moreover, the authors tested a number of cytokines that have a role in immunity, but their role on brain, i.e. on mood is less clear.

The authors should determine whether the change in QOL is a true change (and not linked to a positive or negative reinforcement due to the administration of IVIg per se).

It is likely that a depression questonnaire could be helpful.

Also, the authors should consider how the administration of IVIg leads to the apparent feeling or freedom and safety thus allowing 8and leading into) a better QOL.

Finally, all the immunologic changes are somehow expected, but how do they impact on patients moddo and functionality? any relationship with psychological behaviours?

We appreciate the interest in the paper. It is clear that QOL encompasses a number of factors as identified by the reviewer. To assess QOL we utilized a validated instrument. This was not to imply there were no limitations to the approach but one that appeared effective in addressing our primary target, the “wear-off” effect. There are many reports of cytokine changes affecting behavior and mood through the CNS. We have modified the manuscript discussion section to address this issue and included a relevant reference. Mood questionnaires per se were not part of our approach as they appear more targeted to assessing depression and bipolar function. The SF-12 is a validated health-related quality-of-life questionnaire consisting of questions that measure health domains to assess both physical and mental health. As the majority of the patients were on long-standing, regular IVIG infusions , it is not clear how we could assess “apparent feeling or freedom and safety..leading to a better QOL."

The revised paragraph is pasted here:

This study aimed to identify associations between cytokine changes and variation in QOL through the IVIG cycle. Many of the cytokines monitored have been implicated in altering CNS function and mood, and as a result, may affect QOL determinations (19). Despite identifying quite large immediate cytokines changes, we did not find associations between these changes and the reported QOL improvements. There are multiple likely reasons for this. First, the non-naive group, which made up most of the study population, experienced lower magnitude and less significant cytokine level changes than the naive group, so there were less dramatic changes available for comparison. Second, the low number of enrolled subjects restricts the power of the study to identify subtle correlations missed here. Third, it is possible that timing of this relationship differs from that designed in the study in that additional analyte changes may be occurring between the infusion and day 7. Fourth, the relationship between cytokine fluctuations in the blood and alterations in the CSF is not entirely clear. It is conceivable that changes in the CNS such as blood-brain barrier alterations, increases or decreases in glymphatic flow, or changes in tissue cytokine levels may have minimal impact on serum levels, particularly when dramatic shifts in analyte levels seem to be simultaneously occurring in other parts of the body. Given the demonstrated role of LPS in altering CNS glymphatic function (20), the known role of low level chronic LPS exposure in CVID (21), and the ability of IVIG to reduce LPS levels in the blood, a corresponding change in CNS conditions may be expected in patients that suffer from chronic LPS exposure. Considering these aspects, it is difficult to rule out that cytokine changes directly or indirectly induced in the CNS do in fact correlate with the reported QOL changes. Innovative approaches to monitoring CNS physiology would be helpful to further understand these relationships.

Reviewer #2: This is an interesting paper, but a bit more data and interpretation could help the reader assess the applicability and clinical significance of the conclusions to individual patients. The data and comments requested below should be available already, but should be added to the presentation. It is probably not feasible at this time, but it would have been informative to have weekly data points throughout each infusion-infusion cycle, rather than just comparing the pre-infusion and 7-day time points.

Answers to a few questions would help the readers focus on the results:

1. Were pre-infusion meds such as NSAIDS, aspirin or steroids allowed, or used in any patients ? Were there any clinically significant infusion reactions and did these lead to results different from infusions with no reactions ?

Preinfusion medications were administered in 8 of the subjects. Four received acetaminophen only. Two received acetaminophen and diphenhydramine. One received hydrocortisone 100mg, and one received both hydrocortisone 100mg and acetaminophen. There were no significant infusion reactions. Pretreatment did not significantly impact any of the significant findings including VAS, fatigue, general health, Treg percentage, or any of the cytokines changes from pre-infusion to either post-infusion or day 7.

Three additional supplemental figures are now included that show the relationship between pretreatment on the specified outcomes: change in VAS pre-post and pre-day7, change in fatigue score pre-post and pre-day7, change in general health pre-post and pre-day7, Treg pre-post and pre-day7s, and change in cytokine levels pre-post and pre-day7.

2.Were any patients on continuous prophylactic or treatment antibiotics ?

Information regarding antibiotic use was not collected for the group. The authors no longer have access to the patient's charts as the study is closed and they are no longer at the institution. Anecdotally, patients treated at this center at the time of the study were infrequently on prophylactic antibiotics.

3. Please add to table 1 data on the duration of IVIG treatment and/or number of infusions preceding enrollment for each subject.

We appreciate this excellent suggestion, as it would be nice to determine if there is a correlation between duration of IVIG treatment and cytokine changes within the experienced IVIG user group. Unfortunately, the precise length of time that each subject was on IVIG prior to enrollment was not recorded or extracted from the medical record. Only whether the subject started IVIG during the trial was recorded. It is therefore not possible to add this information to the table or to perform an analysis of the relationship between duration of IVIG and cytokine changes in this group.

4. The authors should clarify if their use of the term "VAS" score refers only to their 100 point "thermometer" and not to the 5 point boxed choices on the modified SF-12.

The visual analogue scale (VAS) score refers to the “thermometer.” This fact has been clarified in the text describing the questionnaires. “The VAS is a single question where subjects rate their health on a scale of 0 to 100.”

5. In fig 2B, the authors should indicate the number of values for each response category (1-4), indicate whether any subject ever reported a value of 5, and comment on the consistency of individual subjects' responses over the multiple intervals represented in the figure.

Thank you for this helpful suggestion. We looked more deeply into the trends of responses to this question and found that reporting worse health over the past week was more common on infusion day than on day 7. We added additional text to the result section indicating as such. We also now provide a new supplemental figure (fig S2C) that shows the reported score for each individual over the course of the study. To directly answer your questions, no subject responded with a 5 for the weekly health question, and the numbers of each response were as follows: “1” = 6, “2” = 21, “3” = 62, “4” = 12, “NA” = 5. These numbers were added to figure 2B.

To aid in interpreting the results, the authors should discuss whether there is any data on the clinical relevance of the change in Tregs from 2.28 % to 2.42% portrayed in Fig 3, and discuss the minimal clinically important difference in this measurement, as well as in the cytokine values in fig 4. Is there any data available on the range of results for normal controls in their laboratory?

The median Treg% increased as described, but some individuals experienced much larger changes and others experienced no change or a decrease. The most we can say regarding these changes within the context of this study is that the observed changes in Treg percentage did not correlate with an improvement in VAS or SF-12 responses; and moreover, the changes are not apparently the result of the observed cytokine changes. Therefore, the increase in Treg percentages dis not appear associated with clinical benefit. The reference range for Treg in the healthy local population is 1.8 to 8.3% of CD4+ T cells. There is no reference range available for the cytokine measurements, as the assay was performed solely on a research basis for this study. The clinical significance of the detected cytokine changes is not clear as they did not correlate with any measured clinical outcome. This conclusion may reflect the fact that the largest group of recipients were on an established regimen of infusions.

In addition, different symbols or colors should be used for IVIG-naive vs experienced subjects. 

In the original figure 4, circles denote the IVIG-naive subjects and triangles denote the IVIG-experienced subjects. Red was used to denote statistically significant results. 

The authors should also discuss the degree of variability vs consistency in Treg responses in different infusion cycles in individual subjects (fig S3, and S4-S5)

There was substantial variability among individuals from cycle to cycle. Five subjects experienced an increase in Tregs every cycle, 8 experienced an increase on 2 cycles, and 5 experienced an increase during only 1 cycle. No subjects failed to experience a Treg increase during at least 1 cycle. This information was added to the section describing the Treg results.

Fig 5 could also be split into two figures- one showing the IVIG-naive subjects and one showing the experienced subjects.

Thank you for this suggestion. The figure has now been separated into 2 figures. It is clear from the figure containing the naïve-subject trends that they have uniform cytokine responses, and we agree that it is better to present the figures in this way.

The authors conclude that the changes in IL-25 they report may partially explain "the clinical efficacy of IVIG in allergic disease", citing the over-20 year-old reference #21 in regard to this "efficacy". What is the evidence that the decrement in IL-25 they report is clinically significant? Given the lack of substantiation of the efficacy of IVIG in most atopic diseases, and the facts that the FDA has not approved IVIG for any atopic disease, and that the AAAAI review of uses of IVIG does not recommend it for atopic disease, this conclusion appears to be an overstatement rather than a "take-home message" as it seems currently stated. The authors should probably substitute the latest AAAAI recommendations for IgG therapy (Perez et al JACI 2017) for the outdated reference 1 they now cite.

We acknowledge the controversial nature of this statement and have deleted it and the reference.

---

## [Decision Letter · Decision Letter 1]

9 Mar 2022

Fluctuations in quality of life and immune responses during intravenous immunoglobulin infusion cycles

PONE-D-21-33136R1

Dear Dr. Abbott,

We’re pleased to inform you that your manuscript has been judged scientifically suitable for publication and will be formally accepted for publication once it meets all outstanding technical requirements.

Kind regards,

Jagadeesh Bayry, DVM, PhD, HDR

Academic Editor

PLOS ONE

Additional Editor Comments (optional):

Reviewers' comments:

Reviewer's Responses to Questions

**Comments to the Author**

1. If the authors have adequately addressed your comments raised in a previous round of review and you feel that this manuscript is now acceptable for publication, you may indicate that here to bypass the “Comments to the Author” section, enter your conflict of interest statement in the “Confidential to Editor” section, and submit your "Accept" recommendation.

Reviewer #1: All comments have been addressed

2. Is the manuscript technically sound, and do the data support the conclusions?

Reviewer #1: Yes

3. Has the statistical analysis been performed appropriately and rigorously? 

Reviewer #1: No

4. Have the authors made all data underlying the findings in their manuscript fully available?

Reviewer #1: Yes

5. Is the manuscript presented in an intelligible fashion and written in standard English?

Reviewer #1: Yes

6. Review Comments to the Author

Reviewer #1: The authors properly responded to all raised comments and the paper is now suitable for publication

7. PLOS authors have the option to publish the peer review history of their article (what does this mean?). If published, this will include your full peer review and any attached files.

Reviewer #1: No

---

## [Editor Report · Acceptance letter]

14 Mar 2022

PONE-D-21-33136R1 

Fluctuations in quality of life and immune responses during intravenous immunoglobulin infusion cycles 

Dear Dr. Abbott:

I'm pleased to inform you that your manuscript has been deemed suitable for publication in PLOS ONE. Congratulations! Your manuscript is now with our production department. 

Kind regards, 

on behalf of

Dr. Jagadeesh Bayry 

Academic Editor

PLOS ONE